# CSPG4-Specific CAR T Cells for High-Risk Childhood B Cell Precursor Leukemia

**DOI:** 10.3390/ijms20112764

**Published:** 2019-06-05

**Authors:** Dennis C. Harrer, Gerold Schuler, Jan Dörrie, Niels Schaft

**Affiliations:** Department of Dermatology, Universtitätsklinikum Erlangen, Friedrich-Alexander-Universität Erlangen-Nürnberg, Hartmannstraße 14, 91052 Erlangen, Germany; dennis.harrer@uk-erlangen.de (D.C.H.); gerold.schuler@uk-erlangen.de (G.S.); jan.doerrie@uk-erlangen.de (J.D.)

**Keywords:** CAR T cells, CSPG4, MLL-translocated leukemias, back-up target antigen, high-risk childhood B cell precursor leukemia

## Abstract

The advent of CD19-specific chimeric antigen receptor (CAR) T cells has proven to be a powerful asset in the arsenal of cancer immunotherapy of acute lymphoblastic leukemia and certain B cell lymphomas. However, a sizable portion of patients treated with CD19-CAR T cells relapse with CD19-negative cancer cells, necessitating the quest for back-up antigens. Chondroitin sulfate proteoglycan 4 (CSPG4) expression has been reported on leukemic blasts bearing the ill-fated *MLL* 11q23 rearrangement. We aimed at exploring the use of CSPG4-specific CAR T cells against mixed-lineage leukemia (MLL)-rearranged leukemic blasts, using the precursor B cell leukemia cell line KOPN8 (MLL–MLLT1 translocation) as a model. First, we confirmed CSPG4 expression on KOPN8 cells. Bulk T cells electroporated with mRNA encoding a CSPG4-specific CAR upregulated activation markers and secreted the Th1 cytokines TNF and IFNγ in an antigen-specific manner upon co-culture with KOPN8 cells. More importantly, CSPG4-specific CAR T cells evinced specific degranulation towards KOPN8 cells and specifically lysed KOPN8 target cells in chromium lysis experiments. CSPG4 is a well-established CAR target in cutaneous melanoma. Here, we provide proof-of-principle data for the use of CSPG4-specific CAR T cells against MLL-translocated leukemias.

## 1. Introduction

CD19-specific chimeric antigen receptor (CAR) T cells have mediated substantial tumor regressions in patients with relapsed or refractory B-cell malignancies, resulting in their recent FDA approval for acute lymphoblastic leukemia (ALL) and certain non-Hodgkin lymphomas [1]. Impressive complete remission rates above 80% in B-ALL and around 40% in diffuse large B cell lymphoma were achieved by a single infusion of autologous CD19-CAR T cells [2,3,4]. 

CARs are composed of an extracellular single chain Fv detecting membrane-bound antigens and an intracellular CD3ζ activation motif linked in *cis* with a co-stimulatory domain, such as CD28 or 4-1BB [5]. To maximize durable response rates, it is warranted to select target antigens that exhibit consistent expression on malignant cells without inducing serious on-target/off-tumor toxicities. Whereas depletion of conventional CD19-positive B cells can be easily compensated for by immunoglobulin substitution therapy, CD19 shut down represents a major immune escape mechanism compromising ongoing responses to CD19-CAR T cell therapy [6]. Thus, albeit initial CD19 positivity and stringent complete responses after CD19-CAR T cell infusion, a sizable portion of patients suffering from B-ALL relapse with CD19-negative blasts [2,7,8]. CD19 negativity originating from either mutational loss or posttranscriptional editing can be counteracted by using alternative B cell target antigens such as CD22 [9,10]. Recently, CD19 loss predicated on a myeloid lineage switch conferring negativity for all B cell specific antigens has been reported [11,12]. In those cases, B cell mixed-lineage leukemia blasts bearing CD19 and CD22 morphed into CD19/CD22 double-negative myeloid blasts. Hence, creating a diversified portfolio of back-up antigens for CD19 beyond B cell antigens is an urgent need.

Chondroitin sulfate proteoglycan 4 (CSPG4), formerly denoted as melanoma-associated chondroitin-sulfate proteoglycan (MCSP) or high-molecular-weight melanoma-associated antigen (HMW-MAA) is a heavily glycosylated transmembrane protein [13], which is overexpressed in a variety of prognostically unfavorable entities, such as melanoma [14], glioma [15], and triple-negative breast cancer [16]. Moreover, CSPG4 has been found on the surface of MLL-rearranged leukemia cells [17,18,19,20,21], a form that accounts for around 10% of all leukemias [22]. The MLL protein, encoded by the *KMT2A* gene, is a regulator of gene expression owing to its intrinsic methyltransferase activity [23]. In MLL leukemias, the *KMT2A* gene is disrupted ensuing chromosomal translocation [22]. This results in abrogated MLL protein expression and subsequent global demethylation, which provides a possible rationale for the correlation of CSPG4 upregulation and MLL rearrangement. In light of high relapse frequencies and a significantly reduced overall survival associated with MLL leukemia [22], novel treatment strategies are highly required. Hence, antigen-specific targeting of CSPG4 has garnered increasing interest, especially given an elevated resistance of MLL cells to standard chemotherapy [24].

Initial efforts to specifically attack CSPG4 on cancer cells have encompassed monoclonal antibodies and immunotoxins [25]. A variety of antibodies, such as monoclonal antibody 9.2.27 against melanoma [26], monoclonal antibody 225.28 against breast cancer [16], TP41.2 against mesothelioma [27], as well as a single-chain Fv construct, scFv–Fc21 [28], have been successfully employed to stunt tumor progression in animal models, which is largely ascribed to blockade of CSPG4-mediated pro survival signals [16,25,26,29]. Besides, antigen-specific direct cytolysis of CSPG4-expressing targets could be induced using immunotoxins, merging a CSPG4 scFv with a cytotoxic protein, e.g., Exotoxin A [30,31] or microtubule-associated protein tau [32].

Other strategies to specifically eliminate CSPG4-positive targets include fusion proteins linking a CSPG4 binding domain to soluble TRAIL (TNF-related apoptosis-inducing ligand) agonists to initiate cell death upon CSPG4 binding through the extrinsic apoptosis pathway [33]. Regarding MLL leukemia, data targeting CSPG4 are scant. A single study evaluating a CSPG4-specific monoclonal antibody did not show any significant impact on MLL cells in a NOD/SCID model [34]. To date, no further clinical or preclinical data assessing CSPG4 as a target antigen in B-ALL, especially MLL B-ALL, have been reported.

Consistently, all studies involving CSPG4-CAR T cells have been focused on solid tumors, and no data concerning leukemia have been published. T cells, retrovirally transduced with a CSPG4-specific CAR, exerted potent cytotoxicity in various CSPG4-expressing tumors, such as melanoma, breast cancer, mesothelioma, glioblastoma, and osteosarcoma [35,36], in animal models. Additionally, intracranial application of CSPG4-CAR T cells in a murine model of glioblastoma imposed efficient tumor control [37]. A potential caveat of CSPG4-CAR T cell therapy is posed by the fact that CSPG4 expression is not exclusively restricted to malignant cells. Among others, CSPG4 has been detected on activated pericytes [38,39] and, to a far lower extent, on smooth muscle cells [40]. In order to obviate concerns about potential on-target/off-tumor toxicities, we have previously demonstrated that transient transfection of T cells with CSPG4-CARs using mRNA electroporation might be an effective and safe tool in cancer immunotherapy [41,42]. 

The aim of the current study was to extrapolate our experience in targeting melanoma cells with CSPG4-CAR T cells to CSPG4-postiive MLL leukemias using MLL1-MLLT1-translocated KOPN8 B-ALL cells as target cells. We wish to raise initial awareness for CSPG4 as a possible back-up antigen in B-ALL, with special emphasis on MLL-rearranged leukemias. To our knowledge, this is the first study targeting leukemia cells.

## 2. Results

### 2.1. KOPN8 Leukemia Cells Express CSPG4

Mixed-lineage leukemias (MLL) are characterized by specific translocations involving the *MLL* gene on chromosome 11. The target cell line employed in this study, KOPN8, was derived from a patient with precursor B-ALL and harbors the MLL–MLLT1 translocation connecting chromosomes 11 and 19 (t(11;19)) [43]. It has been reported that CSPG4 is expressed on the surface of MLL-rearranged leukemias [17,18,19,20,21]. In order to examine the targetability of KOPN8 cells with CSPG4-CAR T cells, we first analyzed CSPG4 expression using the 9.2.27 anti-CSPG4 antibody, which can bind to diverse isoforms of CSPG4 originating from a variety of different glycosylation patterns. Importantly, the single-chain Fv of the CSPG4-CAR utilized in this study was derived from this antibody, suggesting that this antibody is particularly appropriate for probing the susceptibility to CSPG4-CAR T cells. Uniform expression of CSPG4 was detected on the surface of KOPN8 cells (Figure 1). CSPG4-negative T2.A1 cells did not display a difference between isotype staining and anti-CSPG4 staining. Human melanoma cells A375M, which evince a strong CSPG4 expression, served as positive control for CSPG4-CAR T cell activity throughout this study. 

Collectively, uniform CSPG4 expression on KOPN8 leukemia cells suggested possible targetability by CSPG4-CAR T cells. 

### 2.2. CSPG4-CAR T Cells are Activated by KOPN8 Leukemia Cells in An Antigen-Specific Fashion

Having confirmed the presence of our target antigen, we next sought to evaluate whether CSPG4-CAR T cells could antigen-specifically target KOPN8 leukemia cells. To this end, bulk T cells from monocyte-depleted, healthy donor-derived PBMCs were selectively expanded with OKT-3 and IL-2 for 10 days. This resulted in a homogenous CD3^+^ T-cell population, with a dominant CD8^+^ T-cell fraction (Appendix A). Next, these T cells were electroporated with mRNA encoding a CSPG4-specific, CD28-co-stimulated, second-generation CAR (Figure 2a). Mock (no RNA)-electroporated T cells and T cells transfected with a carcinoembryonic antigen (CEA)-specific control CAR served as controls for CSPG4-CAR specificity. Surface staining for CEA on target cell lines revealed a very weak expression of CEA on KOPN8 leukemia cells and no detectable expression on T2.A1 and A375M cells in comparison to CEA-positive KATO III gastric carcinoma cells (Appendix A). Capitalizing on the pronounced difference in antigen expression levels (CSPG4 high and CEA very low) on KOPN8 cells, CEA-CAR was used to control for unspecific tonic CAR signaling, in order to confirm antigen-specific reactivity of CSPG4-CAR against KOPN8 cells. Upon electroporation, similar expression of CSPG4 and CEA CARs was observed, with transfection rates around 86% (Figure 2b). To determine whether CSPG4-CAR T cells can react to KOPN8 leukemia cells in an antigen-specific manner, mock T cells, control CEA-CAR T cells, and CSPG4-CAR T cells were co-incubated with T2.A1, KOPN8, and A375M cells at a 1:1 ratio, 24 h after electroporation. After a co-culture period of 4 h, upregulation of the activation marker CD69 was analyzed (Figure 2c). Additionally, CD25 expression was assayed following 20 h of co-culture (Figure 2d).

Mock T cells did not upregulate CD69 or CD25 after co-incubation with KOPN8 leukemia cells (Figure 2c,d). CSPG4-CAR T cells evinced a significantly higher CD69 and CD25 upregulation compared to control CEA-CAR T cells following co-culture with KOPN8 leukemia cells, confirming antigen-specific activation mediated via CSPG4 CAR. Moreover, CSPG4-CAR T cells exhibited a significantly higher CD69 and CD25 expression upon co-culture with KOPN8 leukemia cells as compared to co-incubation with CSPG4-negative T2.A1 cells, further corroborating antigen-specific activity of CSPG4-CAR T cells in response to KOPN8 leukemia cells. Human melanoma cells (A375M) served as a positive control for the stimulation of CSPG4-CAR T cells and correspondingly induced upregulation of CD25 and CD69 expression on CSPG4-CAR T cells. Control CEA-specific CAR T cells slightly upregulated CD25 beyond background in response to melanoma cells but not to leukemia cells, which was unexpected considering the absence of CEA on A375M cells. A possible explanation for this conundrum might derive from a potential cross-reactivity of the CEA-CAR with surface molecules on melanoma cells that share antigenic sites of CEA [44]. 

In sum, these data indicated that KOPN8 leukemia cells can be antigen-specifically targeted by CSPG4-CAR T cells.

### 2.3. CSPG4-CAR T Cells Secrete Th1 Cytokines Following Co-Culture with KOPN8 Leukemia Cells

Next, we examined the cytokine secretion profile of CSPG4-CAR T cells in response to KOPN8 leukemia cells. Receptor-transfected T cells were stimulated with T2.A1 cells, KOPN8 cells, and A375M melanoma cells, and the secreted cytokines were quantified after overnight culture. CSPG4-CAR T cells produced significantly more IFNγ than control CEA-CAR T cells and mock T cells upon co-culture with KOPN8 leukemia cells (Figure 3). Additionally, CSPG4-CAR T cells did not secrete IFNγ upon co-incubation with T2.A1 cells, confirming antigen-specific IFNγ secretion towards CSPG4-positive KOPN8 leukemia cells (Figure 3). Moreover, CSPG4-CAR T cells exhibited IFNγ production upon co-culture with human melanoma cells A375M, serving as positive controls (Figure 3). Control CEA-CAR T cell IFNγ production was absent in response to T2.A1 cells but could be found upon co-culture with A375M melanoma cells. Regarding TNF secretion, a similar pattern was observed. In comparison to mock and CEA control CAR T cells, CSPG4-CAR T cells secreted significantly more TNF upon co-culture with KOPN8 leukemia cells (Figure 3). No tonic TNF production by CAR T cells was detected following co-culture with T2.A1 cells (Figure 3). As expected, CSPG4-expressing A375M melanoma cells evoked TNF production by CSPG4-CAR T cells (Figure 3). Whereas IFNγ and TNF are canonical pro-inflammatory cytokines, IL-2 has attained a more dichotomous role in cancer immunotherapy, with T cell stimulatory capacities on one hand and immunosuppressive characteristics on the other hand. Immunosuppression is largely mediated by IL-2 uptake by T regulatory cells, which are subsequently activated and promote immune evasion. Strikingly, while antigen-specific IL-2 production was detected following co-culture with A375M cells, hardly any IL-2 secretion by CSPG4-CAR T cells was observed upon stimulation with KOPN8 leukemia cells (Figure 3). Finally, we screened for secretion of the Th2 cytokine IL-4, which is considered to have immunosuppressive properties. No relevant IL-4 production by CSPG4-CAR T cells was measured after stimulation with KOPN8 leukemia cells and A375M target cells (Appendix A). In aggregate, these results proved that CSPG4-CAR T cells evince an antigen-specific secretion pattern of Th1 cytokines in response to KOPN8 leukemia cells, without IL-2 and IL-4 production.

### 2.4. CSPG4-CAR T Cells Specifically Lyse Leukemia Cells

To test the cytolytic activity of CSPG4-CAR T cells, CEA control T cell, and mock T cells toward KOPN8 leukemia cells, we first performed a 4 h degranulation assay analyzing CD107a upregulation on CD8^+^ T cells after co-incubation with T2.A1, KOPN8, and A375M cells. While neither mock-transfected nor CEA control CAR-expressing CD8^+^ T cells displayed any relevant degranulation towards KOPN8 leukemia cells, CD8^+^ T cells equipped with the CSPG4-CAR exhibited antigen-specific degranulation in response to KOPN8 cells (Figure 4a). Control CEA-CAR T cells showcased a slightly elevated degranulation against melanoma cells in comparison to background degranulation against T2.A1 cells. Thus, CSPG4-CAR T cells displayed antigen-specific cytotoxicity towards KOPN8 leukemia cells. Nevertheless, the most important feature of CAR-T-cell therapy is antigen-specific tumor destruction. Hence, we sought to corroborate the cytotoxic potential of CSPG4-CAR T cells toward KOPN8 leukemia cells by a 4–6 h chromium lysis assay. KOPN8 and T2.A1 cells were used as targets. Mock T cells, CEA-CAR T cells, and CSPG4-CAR T cells demonstrated equal background lysis on T2.A1 cells (Figure 4b). Mock T cells did not induce any lysis in KOPN8 cells beyond background (Figure 4c). CSPG4-CAR T cells antigen-specifically lysed KOPN8 leukemia cells even at low effector-to-target ratios (Figure 4c). Statistical significance for antigen-specific lysis in comparison to the CEA-specific control CAR was achieved at the 20:1 effector-to-target ratio and the 2:1 effector-to-target ratio. CEA-CAR T cells displayed an incrementing cytotoxicity from low to high effector-to-target ratios, presumably reflecting both the concomitant increase in overall tonic CAR signaling with higher effector-to-target ratios and an increasing on-target response against KOPN8 cells, which display a very low expression of CEA (Appendix A). In sum, CSPG4-CAR T cells revealed a high cytotoxic potential as reflected in antigen-specific degranulation and antigen-specific lysis against KOPN8 leukemia cells. Concerning toxicity, no excessive activity against CSPG4-negative targets was observed. 

## 3. Discussion

In the present study, we examined the potential of CSPG4 as a novel target antigen for CAR T cell therapy of MLL B-ALL using MLL–MLLT1 rearranged B cell precursor leukemia cells as targets. We transfected bulk T cells with a CSPG4-specific CAR and assayed several canonical T cell effector functions upon co-culture with CSGP4-expressing KOPN8 leukemia cells. CSPG4-CAR T cells exhibited antigen-dependent upregulation of activation markers and antigen-dependent IFNγ production in response to leukemia cells. Importantly, we could demonstrate that CSPG4-CAR T cells killed KOPN8 leukemia cells in an antigen-dependent fashion. This is the first study reporting on CSPG4-CAR T cells in the context of B-ALL, with special emphasis on the ill-fated MLL B-ALL subtypes.

The poster child for CAR T cell therapy of B-ALL has been CD19, which is initially expressed by virtually all B-ALL blasts [5]. Moreover, as part of the B cell receptor, its physiological distribution is exclusively confined to B cells, limiting the potential for on-target/off-tumor toxicity [45]. Given this favorable expression profile, CD19 has emerged as the primary target in B-ALL. Nevertheless, after high initial response rates, many patients relapse with abrogated CD19 expression [2,7,8]. To date, several mechanisms resulting in the shutdown of CD19 expression have been elucidated: (i) alterations of exon 2, compromising CD19 surface localization [46], (ii) alterations of exon 5, resulting in a truncated form of CD19 [6], and (iii) loss of exon 4, eliminating the binding site of the majority of CD19 CARs [6]. The most obvious strategy to counteract these issues is to target an alternative antigen. However, so far, the arsenal of back-up antigens for CD19 in B-ALL is scant. Promising results have been attained by employing CD22-specific CARs in B-ALL patients including those relapsing after CD19-CAR therapy [9,10]. In a phase 1 clinical trial, 11 out of 15 patients, including 5 patients with prior CD19-CAR T cell treatment, achieved a complete remission following infusion of CD22-CAR T cells [9]. The median remission time, however, did not exceed 6 months, which could be largely traced back to the emergence of leukemia cells with reduced CD22 expression, eluding CD22-CAR T cells. Thus, a more comprehensive arsenal of back-up antigens in ALL is required. This is even more evident against the backdrop of a recent study reporting on the successful deployment of CD19-CAR T cells against 11q23-rearranged MLL B-ALL leukemia [11,12], which is associated with a poor prognosis and preponderance in young patients [22]. All seven patients with MLL B-ALL achieved a complete response upon CD19-CAR infusion [11]. Two patients, however, exhibited a rapid relapse with CD19-negative blasts. Molecular and phenotypic analysis of the relapsing blasts revealed a clonal lineage switch from MLL-B-ALL to MLL-AML conferring negativity for the B-lineage markers CD19 and CD22. This highlights a new immune escape mechanism to CD19-CAR T cell therapy and exposes the dependence on B-lineage antigens as a particular Achilles heel in CAR T cell therapy of MLL-B-ALL. Thus, the creation of a diverse antigen portfolio bears particular relevance for MLL B-ALL. The biology of MLL leukemias is governed by the reactivation of stem cell programs transforming hematopoietic stem cells into highly aggressive leukemic blasts [22]. Intensive combinational chemotherapy regimens have only modestly improved survival at the expense of severe therapy-related morbidity [22]. High-risk hematological malignancies are usually consolidated by hematopoietic stem cell transplantation (HSP) capitalizing on the graft-versus-leukemia effect. In MLL leukemia, however, the majority of studies did not reveal any survival benefit imparted by HSP [47]. Given the paucity of effective treatment options, the advent of CD19-CAR T cells has stirred excitement as a new therapeutic modality against MLL B-ALL with an initial CR rate of 100% in a small cohort [11]. However, as mentioned above, the MLL-leukemia inherent plasticity allowed a lineage switch and concomitant evasion from CD19 targeting, resulting in leukemia recurrence in two patients.

Presumably, a major contributor to this plasticity is constituted by the deregulation of promotor methylation and uncontrolled gene activation, which is a repercussion of the 11q23 translocation disrupting the gene coding for the MLL1 methyltransferase [22]. In this study, we intended to exploit this disturbance in gene expression to screen for the presence of CSPG4, which is a well-established CAR-target antigen in melanoma and other solid tumors [35,36,41,42]. The promotor of CSPG4 has been shown to be sensitive to methylation [48], which is reflected by CSPG4 upregulation in several tumor cell lines upon treatment with demethylating agents [48]. Moreover, the global demethylation following MLL1 rearrangement may induce the expression of CSPG4 transcription enhancers or otherwise facilitate its expression. As a model for MLL B-ALL, we selected KOPN8 precursor B cell leukemia cells, characterized by an MLL–MLLT1 translocation (t11;19). Using the anti-CSPG4 9.2.27 antibody, we could demonstrate uniform CSPG4 surface expression by those leukemia cells. The only report on CSPG4-directed targeting of leukemia cell was predicated on the anti-CSPG4 225.28 antibody, the therapeutic efficacy of which against 11q23-rearranged leukemia cells was evaluated in a xenograft mouse model [34]. No detectable impact on survival and tumor growth could be ascertained. Attacking malignant cells with CAR T cells is deemed more powerful than delivering a sole blocking antibody. We have previously shown that T cells equipped with a CAR derived from the 9.2.27 anti-CSPG4 antibody antigen-specifically eliminated melanoma cells [40,41,42]. In the current study, we could show that those CSPG4-CAR T cells antigen-specifically react also toward MLL leukemia cells, providing the only report on CSPG4-CAR T cells against leukemia to date. 

Conspicuously, leukemia cells did not evoke IL-2 secretion from CSPG4-CAR T cells, whereas solid IL-2 secretion could be observed against melanoma cells. Further studies are warranted to elucidate this interesting finding. One possibility is that KOPN8 cells bind or even consume IL-2, although, to our knowledge, no evidence for such a mechanism has been reported. Alternatively, the lower CSPG4 expression on KOPN8 cells compared to A375M cells may have a larger impact on IL-2 secretion by T cells than on TNF and IFNγ secretion. It might be that the threshold of stimulation needed for IL-2 production is higher than that for TNF and IFNγ production. The role of IL-2 in cancer immunotherapy has become a matter of intense debate. In initial studies on adoptive T-cell therapy, IL-2 was widely applied for its stimulatory impact on T cells, promoting growth and survival of tumor-specific T cells administered to cancer patients [49]. T cells grown ex vivo using IL-2 represented the first efficacious T cell product for cancer immunotherapy [49]. Very recently, it was shown that autocrine IL-2 receptor signaling mediated TGF-β resistance in CAR T cells, making these cells more potent in staying active against TGF-β-positive solid tumors [50]. In the last few years, however, IL-2 consumption by CD25-positive regulatory T cells (Tregs) and subsequent inhibition of tumor-specific T cells has garnered increasing attention [51]. Especially in solid tumors, a high prevalence of Tregs in the tumor microenvironment poses a major obstacle to CAR T cell therapy [52]. Regarding leukemia, low-dose IL-2 administered to 84 patients with AML resulted in a pronounced increase of Tregs in the peripheral blood [53]. These regulatory T cells displayed an augmented expression of CTLA-4 and suppressed the effector functions of conventional T cells in vitro. Hence, the absent IL-2 production by CSPG4-CAR T cells in response to leukemia cells might be beneficial to prevent a surge of regulatory T cells. On the flipside, the deficit in IL-2 production might lead to poor proliferation and a compromised long-term persistence of tumor-specific CAR T cells. This might be of special relevance in the setting of permanently transduced T cells, which are destined to proliferate and form tumor-specific memory cells persisting for months or even years. Our approach relies on transiently equipping T cells with CARs using mRNA electroporation. In this scenario, robust proliferation and persistence are not so important, as the transient receptor expression per se necessitates repetitive injections to maintain therapeutic levels of CAR T cells. 

Finally, our results indicate that CSPG4-CAR T cells are capable of specifically eliminating KOPN8 blasts at low effector-to-target ratios. Generally, a major caveat in CAR T cell therapy arises from potential on-target/off-tumor toxicity due to the accidental killing of non-malignant bystander cells co-expressing the target antigen [54]. With respect to CSPG4, no expression has been detected on hematopoietic stem cells, ruling out therapy-associated myeloablation [20]. Concerns about potential on-target/off-tumor toxicity are stirred by CSPG4 presence on smooth muscle cells and pericytes [38,39,40]. Nevertheless, the expression level in those cell types is far inferior compared to that in malignant cells. Of note, we detected a pronounced upregulation of activation markers and cytokine secretion by CEA-CAR T cells against A375M melanoma cells, which were found to be CEA-negative. This might indicate CEA-CAR cross-reactivity with surface molecules sharing epitope similarity with CEA, present on A375M melanoma cells [44]. This further highlights the caution necessary in selecting new target antigens for clinical application. In acknowledgment of the power exerted by CAR T cells, we have previously developed a protocol to generate CSPG4-CAR T cells via mRNA electroporation [42]. Using RNA-transfected CAR T cells harbors the advantage that the receptor expression is temporally restricted, rendering potential off-target and on-target/off-tumor toxicity evanescent as well. Unlike solid tumors, such as melanoma, which are usually located extravascularly, leukemia cells primarily reside in the blood and the bone marrow. Thus, early onset cytokine release syndrome upon infusion of CSPG4-CAR T cells might be more frequent in leukemia than in melanoma [55]. To mitigate safety concerns, an initial use of repetitive injections of RNA-transfected CSPG4-CAR T cells may be beneficial to probe for toxicity. In case of no serious side effects, a switch to permanently transfected CSPG4-CAR T cells may be conceivable.

In aggregate, we wish to highlight the therapeutic potential of CSPG4-CAR T cells as a possible back-up modality for B-ALL relapsing to CD19-CAR T cell therapy, with special emphasis on MLL leukemias, which harbor the potential to undergo a lineage switch and shed all B cell-associated antigens. CSPG-4-CAR T cells generated via mRNA electroporation responded to MLL leukemia cells with antigen-specific upregulation of activation markers, antigen-specific elaboration of Th1 cytokines, and antigen-specific tumor cell lysis. On the basis of these data, we encourage a comprehensive screening of MLL leukemia patients for CSPG4 expression to pave the way for further investigations towards a clinical application of CSPG4-CAR T cells against MLL leukemia. To our knowledge, this is the first study implicating CSPG4-CAR T cells as a potential therapeutic option for MLL leukemia.

## 4. Materials and Methods

### 4.1. Cells and Reagents

Peripheral blood mononuclear cells (PBMCs) were extracted from whole blood obtained from healthy donors following informed consent and approval by the institutional review board (reference number: 251_16 B, by 14 September 2016), using density centrifugation on Lymphoprep (Axis-Shield, Oslo, Norway). Following monocyte depletion via 1 h dish adherence, the remaining non-adherent fraction was cryopreserved and stored at −80 degrees until experimental use.

Target cell lines included the TxB cell hybridoma T2.A1 (kind gift from Prof. Dr. Schulz, Nuremberg), the B-cell precursor leukemia cell line KOPN8 (MLL–MLLT1 translocation; kind gift from Prof. Dr. Slany, Erlangen), and the melanoma cell line A375M (kind gift from Dr. Aarnoudse, Leiden, Netherlands). The Kato III cell line was a kind gift from Dr. Santegoets, Leiden, Netherlands). The cells were maintained in R10 medium containing RPMI 1640 (Lonza, Basel, Switzerland), 2 mM L-glutamine (Lonza), 20 mg/L Gentamicin (Lonza), 2 mM HEPES (PAA, GE healthcare), 2 mM β-mercaptoethanol (Gibco, Life Technologies, Carlsbad, CA, USA), and 10% (*v*/*v*) heat-inactivated fetal calf serum (PAA, GE healthcare, Piscataway, NY, USA).

### 4.2. T Cell Expansion

Non-adherent fractions were thawed and rested for one day in R10 medium. Afterwards, bulk T cells were directly activated with 0.1 µg/mL anti-CD3 antibody OKT3 (Orthoclone OKT3; Jannsen-Cilag, Neuss, Germany). Next, T cells were expanded as previously described [56]. In brief, 1000 IU/mL interleukin-2 (Proleukin; Novartis, Nuremberg, Germany) was added on days 0, 2, 3, 5, and 7. On day 3, the culture density was re-adjusted to 0.2 × 10^6^ cells/mL. On day 7, the total cell culture volume was first doubled and subsequently equally distributed to two culture flasks. After 10 days, T cells were harvested for further experiments.

### 4.3. In Vitro Transcription of mRNA

A second-generation CAR (CSPG4_HL_-CD28/CD3ζ-CAR) directed against CSPG4 (chondroitin sulfate proteoglycan 4) and a second-generation CAR specific for CEA (CEA-CD28/CD3ζ-CAR) were transferred into T cells. The detailed structures of both chimeric antigen receptors were previously reported [42,57]. T7 RNA polymerase (mMESSAGE mMACHINE T7 Ultra kit; Life Technologies, Carlsbad, CA, USA) was employed for in vitro generation of receptor-encoding mRNA, according to the manufacturer’s instructions. Finally, mRNA was purified on RNeasy columns (Qiagen GmbH, Hilden, Germany) in accordance with the manufacturer’s instructions. Before electroporation, mRNA integrity was evaluated by agarose gel electrophoresis.

### 4.4. RNA Electroporation

RNA transfection was performed as detailed elsewhere [58]. In short, following expansion, T cells were washed in OptiMem (Life technologies, Carlsbad, CA, USA,) and transferred to 4 mm-gap electroporation cuvettes (Biolab Products, Bebensee, Germany). The cells were either mock-electroporated (no RNA) or transfected with 15 μg of RNA coding for CSPG4-specific CAR (CSPG4_HL_-CD28-CD3ζ) or for CEA-specific CAR, using a Gene Pulser Xcell (Bio-Rad, Hercules, CA, USA) at 500 V (square wave pulse) for 5 min. After transfection, T cells were cultured in R10 medium.

### 4.5. Flow Cytometry

CSPG4 expression on T2.A1, KOPN8, and A375M cells was detected using an anti-human CSPG4 antibody (BD Biosciences, USA, clone: 9.2.27). IgG2a isotype-stained cells served as controls. Cellular composition after expansion was analyzed on day 10 with anti-CD3 (BD Biosciences, USA, clone: UCHT1) and anti-CD8 (BD Biosciences, USA, clone: SK1) antibodies. IgG1 isotype-stained cells served as controls. CEA expression on T2.A1, KOPN8, A375M, and Kato III cells was analyzed using an anti-human CEA antibody (BD Biosciences, USA, clone: B1.1/CD66). IgG2a isotype-stained cells served as controls.

Surface expression of the introduced receptors was analyzed flow cytometrically 24 h after electroporation. CARs were stained with goat-F(ab’)2 anti-human IgG antibodies (Southern Biotech, Birmingham, AL, USA) directed against the extracellular IgG1 CH2CH3 (Fc-spacer) CAR domain. 

Analysis of T cell activation markers was conducted 24 h after electroporation, upon co-culture (4 h for CD69 and 20 h for CD25) of CSPG4-CAR T cells with T2.A1, KOPN8, and A375M target cells at a 1:1 ratio. T cells, either mock-electroporated or transfected with a CEA-specific CAR, served as controls. T cells were stained with either anti-CD69 (BD Biosciences, USA, clone: FN50) or anti-CD25 antibodies (BD Biosciences, USA, clone: M-A251). The specific mean fluorescence intensity (MFI) was calculated by subtraction of the background MFI obtained with mouse IgG1 isotype control antibodies.

Immunofluorescence was measured using a FACScan cytofluorometer (BD Biosciences, Heidelberg, Germany) equipped with CellQuest software (BD Biosciences). Data were analyzed using FCS Express 5 (De Novo Software, Glendale, CA, USA).

### 4.6. Cytokine Secretion

Cytokine secretion by CSPG4-CAR T cells in response to leukemia cells was assayed as described before [59]. T cells either mock-electroporated or transfected with a CEA-specific CAR served as controls. In short, 24 h after transfection, T cells were stimulated overnight at a 1:1 ratio with T2.A1, KOPN8, and A375M cells. The supernatants were recovered, and the concentrations of the indicated cytokines were quantified using the Th1/Th2 Cytometric Bead Array Kit II (BD Biosciences), in accordance with the manufacturer’s instructions. Immunofluorescence was measured with the FACSCanto (BD Biosciences, Franklin Lakes, NJ, USA) operating with FACSDiva software (BD Biosciences). Data analysis was carried out using FCS Express 5.

### 4.7. Degranulation 

Degranulation of CSPG4-CAR T cells towards leukemia cells was measured using conventional CD107a staining. T cells, either mock-electroporated or transfected with a CEA-specific CAR, served as controls. Twenty-four hours after electroporation, T cells were stimulated at a 1:1 ratio with T2.A1, KOPN8, and A375M target cells. Monensin (eBioscience, San Diego, CA, USA) at a final concentration of 1 µM and an anti-CD107a antibody (BD Biosciences, San Jose, CA, USA, clone: H4A3) were added right at the beginning of co-culture. After 4 h, the cells were stained with an anti-CD8 antibody (BD Biosciences, San Jose, CA, USA, clone: SK1) and analyzed via flow cytometry. Degranulation of CD8-positive T cells was calculated by dividing the portion of CD107a-positive/CD8-positive cells by the portion of CD8-positive cells.

### 4.8. Chromium Release Assay

Specific cytotoxicity of CSPG4-CAR T cells towards leukemia cells was analyzed with a standard 4–6 h ^51^chromium-release assay, 24 h after electroporation, as previously described [59]. T cells, either mock-electroporated or transfected with a CEA-specific CAR, served as controls. In brief, human tumor cell lines T2.A1 and KOPN8 were labeled with 20 µCi of Na_2_^51^CrO^4^/10^6^ (PerkinElmer, Waltham, MA, USA) for 1 h. Next, the targets were plated on 96-well plates and co-cultured with the effectors at the indicated effector-to-target ratios. The supernatants were recovered after 4–6 h, and chromium-release was measured with the Wallac 1450 MicroBeta plus Scintillation Counter (Wallac, Turku, Finnland). The following equation was used to determine the percentage of cytolysis: 100% × [(measured release − background release)]/[(maximum release − background release)].

### 4.9. Figure Preparation and Statistical Analysis

Graphs were created, and statistical analysis was performed using GraphPad Prism, Version 7 (GraphPad Software, San Diego, CA, USA); *p* values were calculated by paired t test, * indicates *p* ≤ 0.05, ** indicates *p* ≤ 0.01, and *** indicates *p* ≤ 0.001.

## Figures and Tables

**Figure 1 ijms-20-02764-f001:**
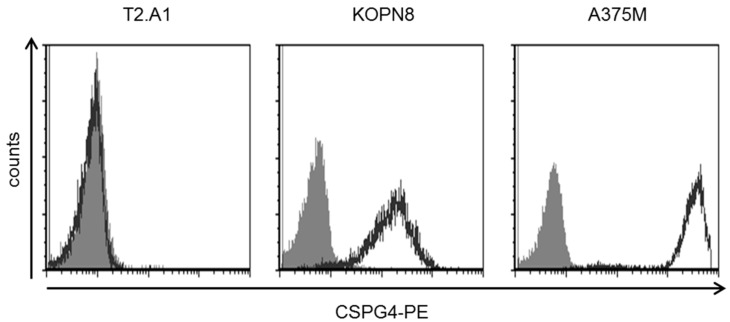
KOPN8 leukemia cells express CSPG4. Surface expression of CSPG4 on KOPN8 cells in comparison to CSG4-negative T2.A1 cells and CSPG4-positive A375M melanoma cells. One representative staining out of three independent experiments is shown.

**Figure 2 ijms-20-02764-f002:**
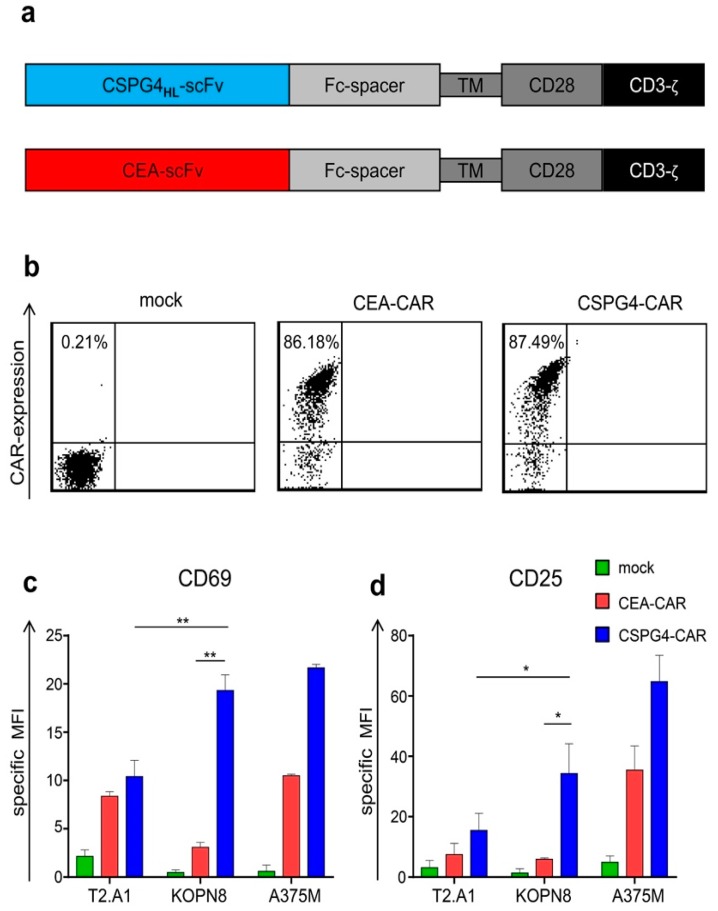
Chondroitin sulfate proteoglycan 4 (CSPG4)-chimeric antigen receptor (CAR) T cells are antigen-specifically activated upon stimulation with KOPN8 leukemia cells. Bulk T cells from healthy donors were selectively expanded using OKT3 and IL-2, as detailed in the Materials and Methods section. After 10 days, these cells were either mock (no RNA)-electroporated or transfected with CAR-encoding mRNA. (**a**) Schematic representation depicting the modular structure of the CSPG4-specific and the carcinoembryonic antigen (CEA)-specific CAR. (**b**) Equal expression of CSPG4 and CEA CARs was confirmed via flow cytometry 24 h after electroporation using a Phycoerythrin (PE)-labeled goat anti-human IgG antibody, binding the fc-part of the CARs. One representative donor out of three independent experiments is shown. (**c**,**d**) Twenty-four hours after electroporation, T cells were co-incubated at a 1:1 ratio with T2.A1 cells, KOPN8 cells, and A375M cells. Mock and CEA-CAR T cells served as negative controls. Upregulation of CD69 (**c**) was analyzed after 4 h of co-culture using anti-human CD69 staining, and upregulation of CD25 (**d**) was detected after 20 h of co-culture using an anti-human CD25 antibody. The specific mean fluorescence intensity (MFI) was calculated by subtraction of the background MFI obtained with isotype antibodies. Data represent geometric means ± SEM from three independent experiments; *p* values were calculated by paired t test, * indicates *p* ≤ 0.05, and ** indicates *p* ≤ 0.01.

**Figure 3 ijms-20-02764-f003:**
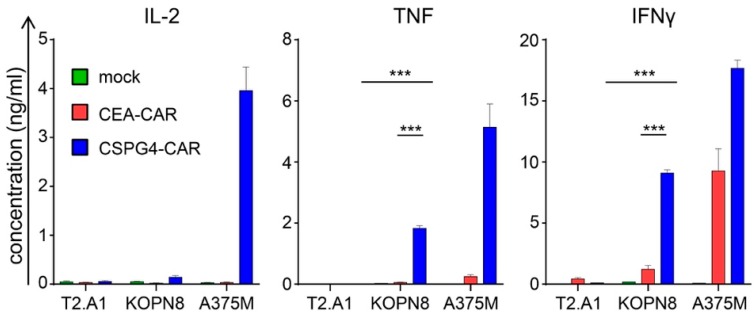
CSPG4-CAR T cells antigen-specifically produce Th1 cytokines in response to stimulation with KOPN8 leukemia cells. The different T cell conditions were generated as mentioned above (Figure 2). Mock (no RNA)-electroporated T cells and CEA-CAR T cells served as negative controls. Twenty-four hours after electroporation, T cells were co-incubated over-night at a 1:1 ratio with T2.A1 cells, KOPN8 cells, and A375M cells. Induced cytokine secretion was quantified in the supernatant with a cytometric bead array (CBA). Concentrations of IL-2, TNF, and IFNγ are depicted [ng/mL]; please note the different scales. Data represent means ± SEM from three independent experiments, *p* values were calculated by paired t test, *** indicates *p* ≤ 0.001.

**Figure 4 ijms-20-02764-f004:**
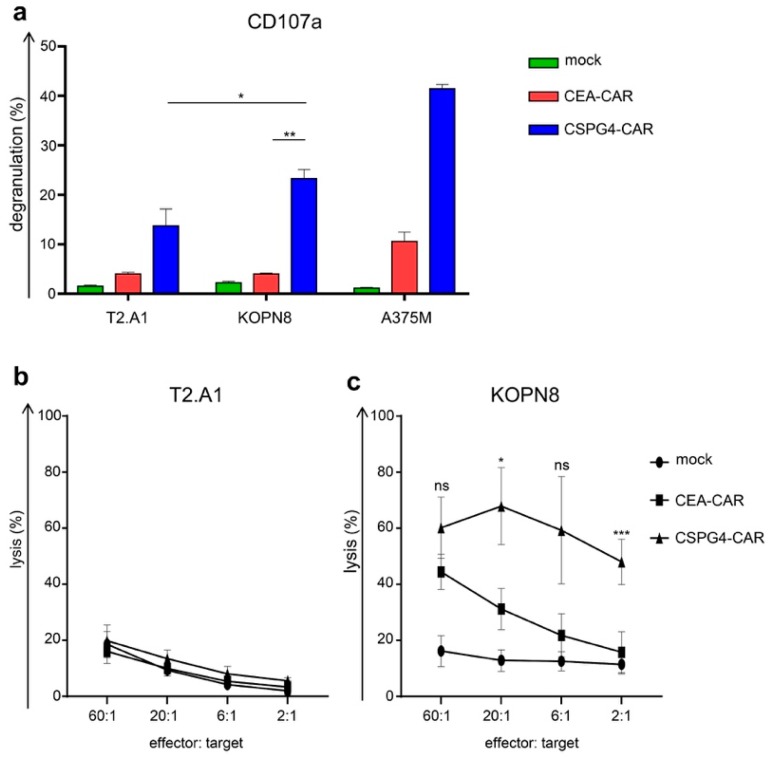
CSPG4-CAR T cells antigen-specifically kill KOPN8 leukemia cells. The different T cell conditions were generated as mentioned above (Figure 2). Mock (no RNA)-electroporated T cells and CEA-CAR T cells served as negative controls. Cytotoxicity towards T2.A1 cells, KOPN-8 cells, and A375M cells was determined 24 h after electroporation. (**a**) Upon 4 h of co-culture with the target cells, degranulation was assayed using CD107a and CD8 staining. Percent degranulation of CD8^+^ T cells was calculated by dividing the portion of CD107a-positive/CD8-positive T cells by the portion of CD8-positive T cells. Data represent means ± SEM from three independent experiments; *p* values were calculated by paired t test, * indicates *p* ≤ 0.05, and ** indicates *p* ≤ 0.01. (**b**,**c**). Lytic capacity towards T2.A1 (**b**) and KOPN8 (**c**) cells was analyzed at the indicated effector-to-target ratios in a standard 4 h chromium lysis assay. Data represent means ± SEM from three independent experiments; *p* values were calculated by paired t test, * indicates *p* ≤ 0.05, ** indicates *p* ≤ 0.01, and *** indicates *p* ≤ 0.001.

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
