# Peer review of "CSPG4-Specific CAR T Cells for High-Risk Childhood B Cell Precursor Leukemia"

_ijms, 2019, doi:10.3390/ijms20112764_

Reviewer 1 Report

A sizable portion of patients with B cell malignancies that are treated with anti-CD19 CAR-T cells relapse due to antigen loss, and this appears to be the case also for anti-CD22 CARs. In a considerable proportion of these patients this loss results from the acquisition of myeloid features accompanied by the loss of both CD19 and CD22. In an attempt to expand the arsenal of targetable antigens in B cell malignancies the authors evaluated a CAR directed at CSPG4, a surface antigen which is frequently expressed by melanomas and several other solid tumors. CSPG4 is also expressed in a subset of B cell tumors that are characterized by a chromosomal translocation which interrupts the methyltransferase activity of the MLL1 gene product, thus activating the CSPG4 promoter. In this manuscript the authors present a proof of concept for the ability of a CSPG4-specific CD28-z CAR to redirect human PBMC-derived T cells against KOPN8, a B cell precursor leukemia cell line which harbors this translocation and expresses CSPG4. As control target cells they employed the CSPG4-negative hybridoma T2-A1 and the CSPG4-postive melanoma cell line A375M and as control for the anti-CSPG4 CAR they used an anti-CEA CAR assembled on an identical molecular backbone.

Using electroporation of in-vitro-transcribed mRNA the authors initially show that the two CARs were properly expressed at the surface of the majority of transfected T cells. They then demonstrate CSPG4-specific activation of the CAR-T cells. This was first evidenced by the upregulation of the activation markers CD69 and CD25, then by increased secretion of IFN-g and TNF-a but not IL-2 (which was only elevated in the presence of the melanoma target cells) and, finally, by antigen-mediated degranulation and target cell lysis.

Overall, the manuscript presents new data on a topic of obvious clinical relevance, is well written and seems to comply with the scope of the IJMS. Yet, there are several major points that need to be addressed by the authors before it can be accepted for publication.

Major points:

1.      Fig. 2 shows the expected upregulation of CD69 and CD25 on the anti-CSPG4 CAR-T cells in the presence of CSPG4-positive targets but, unexpectedly, also on the anti-CEA CAR-T cells in the presence of the T2-A1 and the A375M targets. Since antigen-specificity is a major issue in the CAR-T approach (as well discussed by the authors), a flow cytometry analysis of all three target cells for the expression of CEA is requested, accompanied by an appropriate discussion of the findings obtained in this analysis in light of the activation results.

2.      The reason for the lack of IL-2 secretion following coculture with the leukemia targets, in contrast to massive secretion in the presence of the melanoma targets, is not clear and the authors are requested to propose a mechanistic explanation for these findings.

3.      While discussing the findings on IL-2, the authors mainly refer to potential favorable consequences of this mode of response, but pay little attention to less favorable ones, for example, the role of autocrine IL-2 signaling in resistance to TGF-b suppression, as has recently been demonstrated by the Abken group (Mol. Ther. 2018). The authors are requested to also include in their discussion potential negative effects of the failure to produce IL-2.

4.      In Fig. 4 the authors are requested to include similar data on the killing of the melanoma targets by both CAR-T cells, as done throughout the entire study for this CSPG4-positive control.

Minor point:

5.    First line of the introduction (34) should be removed.

Author Response

Answers to the reviewers

Reviewer 1:

First of all, we thank the reviewer for reading and evaluating our manuscript.

To the raised concerns we have prepared a point-by-point response below (indicated in red):

Comments and Suggestions for Authors:

A sizable portion of patients with B cell malignancies that are treated with anti-CD19 CAR-T cells relapse due to antigen loss, and this appears to be the case also for anti-CD22 CARs. In a considerable proportion of these patients this loss results from the acquisition of myeloid features accompanied by the loss of both CD19 and CD22. In an attempt to expand the arsenal of targetable antigens in B cell malignancies the authors evaluated a CAR directed at CSPG4, a surface antigen which is frequently expressed by melanomas and several other solid tumors. CSPG4 is also expressed in a subset of B cell tumors that are characterized by a chromosomal translocation which interrupts the methyltransferase activity of the MLL1 gene product, thus activating the CSPG4 promoter. In this manuscript the authors present a proof of concept for the ability of a CSPG4-specific CD28-z CAR to redirect human PBMC-derived T cells against KOPN8, a B cell precursor leukemia cell line which harbors this translocation and expresses CSPG4. As control target cells they employed the CSPG4-negative hybridoma T2-A1 and the CSPG4-postive melanoma cell line A375M and as control for the anti-CSPG4 CAR they used an anti-CEA CAR assembled on an identical molecular backbone.

Using electroporation of in-vitro-transcribed mRNA the authors initially show that the two CARs were properly expressed at the surface of the majority of transfected T cells. They then demonstrate CSPG4-specific activation of the CAR-T cells. This was first evidenced by the upregulation of the activation markers CD69 and CD25, then by increased secretion of IFN-g and TNF-a but not IL-2 (which was only elevated in the presence of the melanoma target cells) and, finally, by antigen-mediated degranulation and target cell lysis.

Overall, the manuscript presents new data on a topic of obvious clinical relevance, is well written and seems to comply with the scope of the IJMS. Yet, there are several major points that need to be addressed by the authors before it can be accepted for publication.

Major points:

1.      Fig. 2 shows the expected upregulation of CD69 and CD25 on the anti-CSPG4 CAR-T cells in the presence of CSPG4-positive targets but, unexpectedly, also on the anti-CEA CAR-T cells in the presence of the T2-A1 and the A375M targets. Since antigen-specificity is a major issue in the CAR-T approach (as well discussed by the authors), a flow cytometry analysis of all three target cells for the expression of CEA is requested, accompanied by an appropriate discussion of the findings obtained in this analysis in light of the activation results.

A: We have performed the experiment the reviewer requested and determined CEA expression on the target cell lines T2-A1, KOPN8, A375M, and as positive control on the gastric carcinoma cell line KATO III (now included as supplemental figure 2)(description on page 4 (line 136-141), and page 10 (line 419-421)). We found no CEA expression on T2-A1 and A375M cells, and low CEA expression on KOPN8 cells, the latter probably explaining the data obtained in the cytotoxicity assay (Fig. 4c; see new text on page 6 line 230-236).

Control CEA-specific CAR T cells up-regulated CD25 beyond background in response to melanoma cells but not to leukemia cells, which was unexpected due to the absence of CEA on A375M cells. A possible explanation for this conundrum might derive from a potential cross-reactivity of the CEA-CAR with surface molecules on melanoma cells that share antigenic sites of CEA, which was already described by (Selby et al., Modern Pathology 1992). This is now described in the results section on page 5 (line 172-176) and in the discussion on page 9 (line 350-354). The slight up-regulation of CD69 in response to stimulation with T2-A2 cells we consider to be background activity of the CAR design.

2.      The reason for the lack of IL-2 secretion following coculture with the leukemia targets, in contrast to massive secretion in the presence of the melanoma targets, is not clear and the authors are requested to propose a mechanistic explanation for these findings.

A: The exact mechanism explaining the lower IL-2 secretion upon stimulation with the leukemia target compared to stimulation with the melanoma target is unknown to us, and further studies are warranted to elucidate this interesting finding, however do not fit in the scope of this manuscript. Nevertheless the finding needs to be discussed. One possibility is that KOPN8 cells bind or even consume IL-2, although, to our knowledge, no evidence for such a mechanism has been reported. Alternatively, the lower CSPG4 expression on KOPN8 cells compared to A375M cells has a larger impact on the IL-2 secretion by the T cells than on TNF and IFNγ secretion. It might be that the threshold of stimulation needed for IL-2 production is higher than for TNF and IFNγ production.

These possibilities are now mentioned in the discussion on page 8, line 317-322.

3.      While discussing the findings on IL-2, the authors mainly refer to potential favorable consequences of this mode of response, but pay little attention to less favorable ones, for example, the role of autocrine IL-2 signaling in resistance to TGF-b suppression, as has recently been demonstrated by the Abken group (Mol. Ther. 2018). The authors are requested to also include in their discussion potential negative effects of the failure to produce IL-2.

A: The reviewer is right; the less favorable consequences of a low IL-2 production were not discussed properly in our manuscript. We have now included this in the discussion on page 8 (line 323) and page 9 (line 324-342). 

4.      In Fig. 4 the authors are requested to include similar data on the killing of the melanoma targets by both CAR-T cells, as done throughout the entire study for this CSPG4-positive control.

A: Cytolytic activity of the CSPG4-CAR T cells against the A375M target cell line was already described previously (Uslu et al., Experimental Dermatology, 2016; Krug et al., Cancer Immunology, Immunotherapy, 2015). Inclusion of such an experiment in figure 4 would not strengthen or weaken the point we want to make: i.e. CSPG4 CAR-T cells lyse MLL-rearranged leukemic cells, but only be a confirmation of previously published work. Therefore, we have not included these data. Moreover, from the CEA expression analysis (new supplemental figure 2), we know that A375M does not express CEA and an antigen-specific lysis of A375M cells by CEA-CAR-T cells is unlikely. As seen with the up-regulation of CD25 expression (Fig. 2) it might be that a false positive result is generated due to a potential cross-reactivity of the CEA-CAR with surface molecules that share antigenic sites of CEA (see answer to question 1).  

Minor point:

5.    First line of the introduction (34) should be removed.

A: We are sorry for this mistake. The first line is now removed.

Reviewer 2 Report

The authors explored the use of CSPG4-specific CAR T cells against MLL-rearranged leukemic cell line KOPN8 (MLL-MLLT1 translocation), as a model for B cell leukemia blasts.

Whereas the clinical application of this approach could be very relevant, especially with proposed transduction methods encompassing mRNA transfection, several pitfalls need to be addressed by the authors before the paper publication.

In particular:

In the first figure, authors claim that the presumed mechanism for CSPG4 up-regulation in cells with MLL rearrangement relies on the abrogated methyltransferase activity, dedicating to this assumption the Figure 1a.

I suggest to perform a methylation analysis proving that, or delete the assumption from the paragraph of results.

Although CSPG4 expression has been already demonstrated in B-ALL patients (as even reported in the paper References), for the topic of this paper, could be relevant to show the expression pattern of the antigen in primary samples in respect to e clonal or heterogeneous expression on the leukemic blast cells.

Any animal model to prove CAR- CSPG4 T cell activity has been performed. Thus, the authors should reach at least a high in vitro confidence with the performed experiments proving the relevance of tumor control exert by CAR- CSPG4 T cells. Indeed, the crucial and more relevant data to prove CAR T potency are coming from the cytotoxic assays that they have been carried out. The data showed in Figure 4C are not so impressive, since the lack of statistical significance between control CAR-CEA and CAR- CSPG4 T cell activity versus KOPN8 cells, at all E:T ratio. Authors declare that this could be explained towards overall tonic CAR signaling at higher effector to target ratios. Thus, authors should explain why the same tonic CAR signalling is not present against control cell line T2.A1. Moreover, authors should also include the panel of positive control of CAR- CSPG4 T cell against positive melanoma cell line A375M. If possible, the message of the paper could be further strengthened by the inclusion of the observation whether CAR- CSPG4 T cells are able to recognize and eliminate primary B-ALL blast cells.

Author Response

Answers to the reviewers

Reviewer 2:

First of all, we thank the reviewer for reading and evaluating our manuscript.

To the raised concerns we have prepared a point-by-point response below (indicated in red):

The authors explored the use of CSPG4-specific CAR T cells against MLL-rearranged leukemic cell line KOPN8 (MLL-MLLT1 translocation), as a model for B cell leukemia blasts.

Whereas the clinical application of this approach could be very relevant, especially with proposed transduction methods encompassing mRNA transfection, several pitfalls need to be addressed by the authors before the paper publication.

In particular:

In the first figure, authors claim that the presumed mechanism for CSPG4 up-regulation in cells with MLL rearrangement relies on the abrogated methyltransferase activity, dedicating to this assumption the Figure 1a.

I suggest to perform a methylation analysis proving that, or delete the assumption from the paragraph of results.

A: One possible explanation for the up-regulation of CSPG4 on the KOPN8 cells is the abrogated methyltransferase activity of the MLL1 protein resulting from the MLL1-MLLT1 translocation. This might lead to CSPG4 promotor activation and CSPG4 expression on KOPN8 leukemia cells. CSPG4 promotor activation may either occur upon direct demethylation of this region as outlined in figure 1a, or may be an indirect effect involving e,g, the binding of a transcriptional enhancer up-regulated in response to the global demethylation following MLL1 methyltransferase shutdown (not depicted).

Both possibilities are now described in the results section on page 3 (line 104-110), and in the discussion section on page 8 (line 303-305).

Although CSPG4 expression has been already demonstrated in B-ALL patients (as even reported in the paper References), for the topic of this paper, could be relevant to show the expression pattern of the antigen in primary samples in respect to e clonal or heterogeneous expression on the leukemic blast cells.

A: Unfortunately, we do not have access to primary samples of B-ALL patients and we can not perform this experiment. As the reviewer already indicated, the CSPG4 expression has already been demonstrated in B-ALL patients in the referred papers. As can be seen in Behm et al., Blood, 1996, the expression of CSPG4 is quite homogeneous on leukemic blast cells.

Any animal model to prove CAR- CSPG4 T cell activity has been performed. Thus, the authors should reach at least a high in vitro confidence with the performed experiments proving the relevance of tumor control exert by CAR- CSPG4 T cells. Indeed, the crucial and more relevant data to prove CAR T potency are coming from the cytotoxic assays that they have been carried out. The data showed in Figure 4C are not so impressive, since the lack of statistical significance between control CAR-CEA and CAR- CSPG4 T cell activity versus KOPN8 cells, at all E:T ratio. Authors declare that this could be explained towards overall tonic CAR signaling at higher effector to target ratios. Thus, authors should explain why the same tonic CAR signalling is not present against control cell line T2.A1. Moreover, authors should also include the panel of positive control of CAR- CSPG4 T cell against positive melanoma cell line A375M. If possible, the message of the paper could be further strengthened by the inclusion of the observation whether CAR- CSPG4 T cells are able to recognize and eliminate primary B-ALL blast cells.

A: We apologize for the unclear description of the statistical significance of the lysis in the manuscript (Fig. 4c). CSPG4-CAR T cells antigen-specifically lysed KOPN8 leukemia cells. Statistical significance for antigen-specific lysis in comparison to the CEA-specific control CAR was achieved at the 20:1 effector to target ratio and the 2:1 effector to target ratio (as indicated in figure 4c with * and ***, respectively). This is now described on page 6, line 230-236.

CEA-CAR T cells displayed an incrementing cytotoxicity from low to higher effector to target ratios, presumably reflecting both the concomitant increase in overall tonic CAR signaling with higher effector to target ratios and an increasing on-target response against KOPN8 cells, which display a very low expression of CEA (now included as supplemental Fig. 2). This is now described in the manuscript in the results section on page 4 (line136-141), page 6, (line 230-236) and the materials and methods section on page 10 (line 419-421).

Cytolytic activity of the CSPG4-CAR T cells against the A375M target cell line was already described previously (Uslu et al., Experimental Dermatology, 2016; Krug et al., Cancer Immunology, Immunotherapy, 2015). Inclusion of such an experiment in figure 4 would not strengthen or weaken the point we want to make: i.e. CSPG4 CAR-T cells lyse MLL-rearranged leukemic cells, but only be a confirmation of previously published work. Therefore, we have not included these data.  

As already mentioned above, we do not have access to primary B-ALL blast cells and therefore can not perform a cytotoxicity assay with CART-CSPG4 T cells using these targets.

Round  2

Reviewer 1 Report

The authors provided the requested data/discussion and the manuscript can now be accepted for publication.

Author Response

We thank the reviewer for the positive response

Reviewer 2 Report

Figure 1a has been deleted from the text but not in the figure itself.

I strongly suggest to eliminate assumption from the results section of figure results. Assumption can be included and explained in the discussion. Since no data supporting the Figure 1A has been added to present paper version, I suggest not include it in the publication.

Functional analysis of the CAR vs AML cells could be improved. As the authors declared, the lack of significance at every E/T ratio could be related to a low CEA expression on the KOPN8 target cells. This suggest the CAR-CEA T cells are not the correct negative control in their assay.

Author Response

Reviewer 2:

Figure 1a has been deleted from the text but not in the figure itself.

I strongly suggest to eliminate assumption from the results section of figure results. Assumption can be included and explained in the discussion. Since no data supporting the Figure 1A has been added to present paper version, I suggest not include it in the publication.

A: We have now, as requested by the reviewer, removed figure 1a from the manuscript, including the text describing this part in the results section. The assumption is now only discussed in the discussion section.

Functional analysis of the CAR vs AML cells could be improved. As the authors declared, the lack of significance at every E/T ratio could be related to a low CEA expression on the KOPN8 target cells. This suggest the CAR-CEA T cells are not the correct negative control in their assay.

A: In our lab, we only have CARs specific for three antigens available; namely specific for CSPG4, CEA, and ErbB2 (Her2/neu).

We did not want to use the ErbB2-specific CAR as negative control, because there is literature describing ErbB2 expression on B-ALL cells (e.g. Hicks et al. Cancer Inform. 2013 Aug 27;12:155-73. doi: 10.4137/CIN.S11831, Haen et al. Oncotarget. 2016 Mar 15;7(11):13013-30. doi: 10.18632/oncotarget.7344, Müller et al. Clin Cancer Res. 2003 Aug 15;9(9):3448-53).

That the KOPN8 cell line expresses CEA on its surface, we only found out after doing the staining as requested by the reviewers in reviewing round #1, and came as a surprise to us. Therefore, we can not perform experiments with a correct negative control CAR (i.e. a CAR specific for an antigen not expressed on the KOPN8 target cell line). However, as a surrogate negative control, mock-transfected T cells do not show lysis of the KOPN8 target cells.